# High Power Pulsed LED Driver for Vibration Measurements

**DOI:** 10.3390/s24134103

**Published:** 2024-06-24

**Authors:** Paolo Neri, Gabriele Ciarpi, Bruno Neri

**Affiliations:** 1DICI, Department of Civil and Industrial Engineering, University of Pisa, Largo Lucio Lazzarino 1, 56122 Pisa, Italy; 2DII, Department of Information Engineering, University of Pisa, Via Girolamo Caruso 16, 56122 Pisa, Italy

**Keywords:** pulsed light, high-power LED, custom photodiode, indirect camera pulse width measurement

## Abstract

Vibration measurements pose specific experimental challenges to be faced. In particular, optical methods can be used to obtain full-field vibration information. In this scenario, stereo-camera systems can be developed to obtain 3D displacement measurements. As vibration frequency increases, the common approach is to reduce camera exposure time to avoid blurred images, which can lead to under-exposed images and data loss, as well as issues with the synchronization of the stereo pair. Both of these problems can be solved by using high-intensity light pulses, which can produce high-quality images and guarantee camera synchronization since data is saved by both cameras only during the short-time light pulse. To this extent, high-power Light-Emitting Diodes (LEDs) can be used, but even if the LED itself can have a fast response time, specific electronic drivers are needed to ensure the desired timing of the light pulse. In this paper, a circuit is specifically designed to achieve high-intensity short-time light pulses in the range of 1 µs. A prototype of the designed board was assembled and tested to check its capability to respect the specification. Three different measurement methods are proposed and validated to achieve short-time light pulse measurements: shunt voltage measurement, direct photodiode measurement with a low-cost sensor, and indirect pulse measurement through a low-frame-rate digital camera.

## 1. Introduction

Measuring high frequency vibrations with camera-based methods is a challenging and emerging task in experimental mechanics [1,2,3,4]. As the vibration frequency increases, the corresponding amplitude decreases, thus requiring higher resolution cameras to enhance measurement sensitivity to small displacements [5]. The increase in camera resolution generally corresponds to the decrease in pixel size, resulting in more demanding lighting conditions. High-speed cameras provide the needed sampling frequency [6,7], but they do not fulfill the required resolution specifications since the maximum resolution at full frame rate is in the range of 1 Mp. Thus, many methods based on conventional low-speed cameras were developed, exploiting downsampling approaches [8,9,10,11]. Despite the chosen hardware, if 3D displacements are to be measured, a stereo camera system is to be adopted, thus camera synchronization can become an issue. Additionally, short exposure times are mandatory to avoid blurred images, which further increases the need for efficient and powerful illumination systems. Constant light sources may fail because of insufficient illumination power and excessive overheating. In this scenario, the demand for a proper illumination source is urgent. All the cited issues can indeed be solved by adopting a proper illumination strategy, such that:-Short-time light pulses can be produced to avoid blurred images-High-intensity illumination can be provided by using high-power devices and collimating optics-Cameras synchronization can be relaxed since they will acquire data only when the light pulse is active.

Some solutions are available on the market, but most of them are limited in terms of shortest pulse width and emitted power. Moreover, it is not easy to convert the manufacturer specification into practical guidelines for product selection, since the effectiveness of the illumination systems may be strongly influenced by the specific experimental setup (camera sensor, optics, field of view, etc.), resulting in the risk of erroneous hardware selection. Additionally, the lack of adaptability and customization makes commercial solutions unsuitable for many research applications, which can require a wide range of different operating conditions. Lastly, all the commercial solutions present high costs, which reduces their accessibility for many research centers and impairs the possibility of purchasing different systems for different testing conditions, thus constraining the research activities. In this scenario, some researchers developed custom solutions to provide high-power and short-width light pulses. To this extent, the advancements in Light-Emitting Diodes (LEDs) technology have provided many accessible light sources that can be investigated. A thorough effort was devoted by researchers to reduce the electrical losses of the devices, thus pushing their operation range [12], e.g., by varying the LED chemical composition [13] or the substrate material [14]. The main drawback of such devices is that, during continuous operation, the junction temperature can rise above admissible values, degrading the LED performance, shortening its life, or leading to abrupt failure. For example, it was experimentally proven in [15] that increasing the junction temperature from 45 °C to 125 °C decreased the service time from 65,000 h to 4796 h. To face this issue, junction temperature monitoring systems were developed [16], and innovative and effective cooling systems were studied [17], but they are invasive and can be insufficient for extreme power operations. In this scenario, pulsed operation may be adopted, thus further pushing the maximum current flowing in the LEDs. This requires dedicated electronics and the design of a driving circuit to control the pulsed operation. For example, an LED driver for dimming control was developed [18], but its operation was limited to low frequencies. High frequency or really short pulse widths require special attention in electronic design; otherwise, the required transient response may not be respected. For example, in [19,20], the authors developed a system for imaging flow velocimetry. Even if the developed circuit was effective for the task, it was not flexible in amplitude control since light intensity could only be controlled by changing the power supply. This approach was applied to microfluidic applications in [21] and to Digital Image Correlation (DIC) measurements in [22,23], by using the same electrical scheme as [19], thus having the same limitations. Additionally, the resulting LED performances were assessed by adopting a commercial and expensive photodetector, thus reducing the reproducibility of the approach.

This paper aims to develop an enhanced version of the aforementioned circuits, which follows a similar principle but expands the circuit capabilities in terms of light power analogic control through a DC signal and LED operation measurement through slow sampling rate devices. Additionally, two different low-cost but effective measurement strategies will be presented to allow for light pulse measurements. The first will be based on a cheap photodiode sensor and a custom control circuit. The second will be based on an innovative approach that exploits indirect low-frame-rate camera measurement of the light pulse and a downsampling strategy to retrieve short time pulses. Experimental testing was performed to assess the effectiveness of the proposed LED driver and of the short-time high-power light pulse measurement systems.

## 2. Materials and Methods

The main elements of the LED driving circuit are the LED, a MOSFET used as a switch (SUP7004GZ), a MOSFET driver (MIC4420), a large capacitor (*C*_3_), and a shunt resistance (*R*_S_), as highlighted in Figure 1a with a blue box. On the other hand, the portion of the circuit represented in the orange box is used to estimate the current flowing in the large capacitor *C*_3_ and enhance the driver flexibility by decoupling the supply voltage of the LED (*V*_L_) with respect to the general supply voltage of the driver (*V*_DD_).

The main DC supply voltage *V*_DD_, can be fixed within the range of 4.5–12 V. Subsequently, thanks to the designed control circuit, the LED is supplied with a voltage *V*_L_, which depends on the value of a DC control voltage *V*_C_. The *V*_L_ value can be controlled in the range between *V*_DD_ and *V*_max_ = *V*_TH_ (1 + *R*_3_/*R*_2_) by using the DC control signal *V*_C_ and considering that the internal voltage reference of the LM2585 device is V_TH_ = 1.23 V:(1)$$\left\{\begin{array}{ll} V_{L}=\;V_{\mathrm{DD}} & \mathrm{if}\;V_{C}\;\geq \;V_{\mathrm{TH}}\;+\;\left(V_{\mathrm{TH}}\;-\;V_{\mathrm{DD}}\right)\;R_{2}/R_{3}\\ V_{L}=\;V_{\mathrm{TH}}\;-\;\left(V_{C}\;-\;V_{\mathrm{TH}}\right)R_{3}/R_{2} & \mathrm{if}\;V_{C}\;<\;V_{\mathrm{TH}}\;+\;\left(V_{\mathrm{TH}}\;-\;V_{\mathrm{DD}}\right)\;R_{2}/R_{3} \end{array}\right.$$

The relation between *V*_L_ and *V*_C_ is reported in Figure 1b: the horizontal axis shows the values of *V*c, normalized with respect to *V*_TH_ + (*V*_TH_ − *V*_DD_) *R*_2_/*R*_3_, while the vertical axis shows the values of *V*_L_, normalized with respect to *V*_DD_. The plot clearly shows that *V*_L_ has a linearly decreasing thread in the range 0 ≤ *V*c < *V*_TH_ + (*V*_TH_ − *V*_DD_) *R*_2_/*R*_3_, and a value constantly equal to *V*_DD_ for any *V*_C_ ≥ *V*_TH_ + (*V*_TH_ − *V*_DD_) *R*_2_/*R*_3_. This allows for the setting the LED brightness via software by setting *V*_C_ with simple hardware with analog outputs (such as Arduino boards). Additionally, LED switching can be easily controlled with the *V*_TTL_ voltage, which can be generated by a signal generator or any other trigger source (e.g., camera exposure is active). This voltage is compatible with 3.3 V (0–3.3 V) or 5 V (0–5 V) standards. When *V*_TTL_ is set to high, the LED turns on, and vice versa. The base assumption for the use of the described circuit is that a low-duty cycle signal is used for *V*_TTL_, in the range of 1%. Otherwise, the specification of the large capacitor *C*_3_ and the resistor *R*_4_ needs to be changed since the charging system might not supply enough current to sustain the needs of the LED. Additionally, a higher duty cycle may easily result in LED overheating because of the large current drained in operation. Indeed, a 100% duty cycle would drive the LED in continuous mode operation, and thus the specification of the manufacturer should be strictly respected for both current values and the cooling system. On the other hand, if low duty cycles are used, it is possible to overload the LED with current levels higher than the specification without failure [19] (even if it may result in LED life cycle shortening). This approach ensures that higher light levels can be obtained from each LED, thus obtaining brighter images.

The described circuit was assembled on a prototype board to test its performance. A two-layer matrix board was used to allow for fast and effective circuit adjustments. The circuit prototype was organized and soldered to guarantee accessibility to input and output connectors while also minimizing parasitic resistance and inductance of the connections, which are responsible for speed limitations. The assembled prototype is shown in Figure 2. The total cost of the board and components was less than 25 €. The main components of the circuit in Figure 1a are also indicated on the physical board in Figure 2.

The voltage across the shunt resistor (*V*_S_) is directly proportional to the current flowing through the LED (*I*_LED_). It can be measured with an oscilloscope, considering *I*_LED_ = *V*_S_*/R*_S_. The relation between the shunt measurement and the current flowing in the LED was preliminarily confirmed by using a magnetic field probe on the cable directly connected to the LED. It is worth noting that, since microsecond pulses are to be investigated, a sampling frequency in the range of MHz is needed to monitor the LED current through the shunt resistor. Alternatively, if such equipment is not available, the LED current can be calculated considering the output signal *V*_OUT_ provided by the driving circuit. This voltage, proportional to the current charging *C*_3_ (*I*_C_), can indeed be measured with a low sampling rate:*I*_C_ = *V*_OUT_/*R*_4_(2)

Even if the current is not constant over time, its mean value can be calculated by sampling *V*_OUT_ with a relatively slow frame rate (in the range of 100 Hz, since the circuit time constant is 10 ms) and integrating it over a full period of the signal, obtaining the average charging current *I*_C,a_. Knowing the signal duty cycle (*D*), the mean value of the current flowing through the LED when *V*_TTL_ is high can be computed as:*I*_LED_ = *I*_C,a_/*D*(3)

This method allows obtaining an estimation of the current *I*_LED_ without needing expensive equipment and can be used to verify the proper LED operation instead of measuring *V*_S_, which would require a sampling rate in the range of 10 MHz.

## 3. Experimental Results

### 3.1. Performance Estimation Criteria

Evaluating the LED performance is not trivial, since the fast response time poses severe specifications for any acquisition hardware that can be used. A pulse width in the microsecond range corresponds to a sampling frequency in the range of MHz, thus impairing the use of cameras to directly evaluate the transient behavior. In this scenario, different acquisition methods were investigated in this paper: measurement of the shunt voltage *V*_S_, which is proportional to the current flowing in the LED; direct measurement of the light emission through a low-latency photodiode; indirect measurement of the light emission through a slow camera, and a frequency modulation approach (see Section 3.3). The first method is the easiest to implement since it only requires an oscilloscope and represents an indirect measurement of the light emission through a quantity (*V*_S_) that can be related to the current flowing into the LED and, thus, to the light pulse. The second method represents a direct measurement of the light emission, and it was implemented by assembling a low-cost custom sensor. The last method is the most interesting since it does not require any further equipment but the camera, which is already available in any DIC system and does not need to fulfill strict requirements about acquisition speed. The comparison of the data retrieved with these methods will provide insight into the LED and driver circuit performances.

### 3.2. Direct Photodiode Acquisition

In order to directly acquire the light intensity emitted by the LED, off-the-shelf products are available that guarantee specifications in the MHz range, e.g., the Thorlabs PDA10A photodetector [19], which is rated up to 150 MHz (i.e., 6.7 ns). Nevertheless, because of their relevant cost and lack of adjustability, a low-cost custom solution was developed in this research using a photodiode. Many different photodiodes are available on the market, having negligible cost and ensuring response times in the range of nanoseconds. In particular, the SFH229 (Osram) was selected for this research, with a rise and fall time of 10 ns, which are of the same order of magnitude as the Thorlabs PDA10A. A transresistive amplifier using an operational amplifier was designed to read the current generated by the photodiode, following the scheme reported in Figure 3. In the provided scheme, *V*_P_ is proportional to the photocurrent (*I*_P_) generated by the photodiode, since *V*_P_ = *R*_g_·*I*_P_.

Following this simple scheme, the measurement gain can be set by adjusting the resistor *R*_g_. Nevertheless, the higher *R*_g_, the slower the rise time; thus, a compromise value was chosen in this paper (i.e., 1 kΩ). The readout was routed to an oscilloscope through a BNC cable, with the tension *V*_P_ proportional to the current flowing through the photodiode and, thus, to the light emitted by the LED. The overall cost of the needed components was lower than 4 €. The photodiode was mounted in a 3D printed case, to allow for its stable mounting in front of the LED and to shield the sensitive area with respect to ambient lighting. It is worth noting that, due to the photodiode working principle, when no light is emitted by the LED, the circuit readout will be *V*_P_ = 0 V, while a negative readout will be found if the light is emitted by the LED.

### 3.3. Indirect Camera Acquisition

A method to indirectly measure the LED transient response by adopting a conventional camera and leveraging frequency modulation is proposed in this paper. Two different trigger sources can be used to drive the LED light pulse and the camera exposure. By properly setting the frequencies of these triggers, it is possible to describe the LED transient response even if a slow frame rate camera is available. In particular, if a small difference in the two frequencies is applied, the relative positioning in the time domain between the light pulse and the camera exposure will change along the frames. Additionally, if a proper time delay is imposed on camera exposure, the position of the first frame with respect to the light pulse can be arbitrarily defined. For example, if a delay greater than the camera exposure is chosen, the first frames will be dark, and then the overlap between the light pulse and the exposure will produce images of increasing brightness until the maximum is reached when the light pulse is fully covered by the camera exposure. The image brightness will then decrease when the light pulse passes through the camera exposure. This is clarified with a numerical example in Figure 4a, where camera exposure is schematized with an ideal square wave (blue line), while the light pulse is represented by a transient behavior over time (orange line). Even if the research aims at fast LED operation and short light pulses, the numerical example was set to enhance the plot readability. To this extent, an exposure time of 60 ms and a pulse width of 40 ms were chosen. The camera delay was set to 48 ms, while the camera and LED triggers were set to 10 Hz and 9.5 Hz, respectively. These values can be arbitrarily changed according to the actual experimental needs (see Section 3.3). Figure 4a shows the two signals over time, highlighting how they are not overlapped in frames 1–2. The overlap happens in frames 3–10, and they are completely overlapped in frames 11–13. Subsequently, in frames 14–19, the light pulse passes the camera exposure, and finally, in frames 20–21, they are not overlapped anymore. The effect of this overlap on the image is reported in Figure 4b, where the normalized brightness over time is shown. As can be noted, when the signals are not overlapped, the brightness is zero (frames 1–2 and 20–21); it gradually increases or decreases when the signals are partially overlapped (frames 3–10 and 14–19), and it is constant to the maximum value for the period of complete overlap (frames 11–13).

In the experimental scenario, it is possible to draw the equivalent of Figure 4b by acquiring the blinking LED and computing the average intensity of the pixels corresponding to the emitting surface. Even if it is possible to infer some information about the impulse duration from this plot, some more elaboration is needed to retrieve the actual shape of the light pulse, thus obtaining the equivalent of the orange curve in Figure 4a. Each point in Figure 4b represents the intensity level of a frame acquired by the camera. This intensity is the integral of the light pulse in the time window when the light pulse overlaps with the camera’s exposure (since exposure is a square wave in the range 0–1). Thus, the light pulse shape can be retrieved from the data reported in Figure 4b by deriving the curve. In other words, assuming that the camera exposure is an ideal square wave, it is sufficient to compute the derivative of the curve in Figure 4b to obtain a description of the pulse shape. It is worth noting that this assumption holds in the actual experiments as well, since camera exposure can be considered a logical state between active (1) and inactive (0). Additionally, it is possible to associate each camera frame with the corresponding fictitious pulse time. In other words, since the frequencies of the triggers are not the same, the temporal phase between the two signals will change for each frame by a quantity that can be computed as follows:Δ*t* = 1/*f*_l_ − 1/*f*_c_(4)
where *f*_l_ and *f*_c_ represent light pulse and camera trigger frequencies, respectively. Additionally, Δ*t* is the time shift between the light pulse and the camera exposure, associated with each consecutive frame. The value of Δ*t* can be arbitrarily low, depending on the values of *f*_c_ and *f*_l_. Thus, it is possible to assign to the *i*-th frame acquired by the camera the fictitious time instant *i* × Δ*t*. This leads to the reconstruction of a fictitious time series that represents the light pulse over time, even if strongly downsampled during the acquisition. The results are reported in Figure 5 for the discussed numerical example. Obviously, if Figure 4b is considered, the derivative will be positive on the left side (a light pulse meeting the camera exposure) and negative on the right side (a light pulse passing the camera exposure). Hence, the absolute value of the derivative was plotted in Figure 5, thus showing the light pulse shape twice. In this numerical example, the simulated pulse shape is known (i.e., the orange line in Figure 4a), and it was drawn in Figure 5 with a dashed line. On the other hand, the solid line represents the light pulse reconstructed with the proposed indirect method. The markers highlight the discrete frame instants. It is worth noting that the difference between the triggers in this example was chosen relatively high to enhance the image readability. Thus, the pulse reconstruction is coarse. In an experimental scenario, the temporal resolution of the light pulse reconstruction can be chosen arbitrarily fine by simply reducing the difference between *f*_l_ and *f*_c_, thus providing an interesting insight into the pulse shape. The described procedure does not require any additional hardware besides the LED, the driver, and a camera with a trigger option without any special constraint on its specification.

### 3.4. Experimental Results

The LED investigated in this paper was a Luminus PT-121-G-L11-MPK, rated for 4.8 V and 30 A for continuous operation [24] (provided that high-performance cooling is guaranteed to the LED). The measurements were repeated for different durations of the light pulse. The aim was twofold: assessing the fastest light pulse that could be generated by the system and assessing the LED response time, i.e., the pulse width needed to reach the maximum lighting intensity. The measurements obtained with the photodiode were analyzed first: the signal from the trigger source (*V*_TTL_), the shunt resistor (*V*_S_), and the photodiode (*V*_P_) were compared for the pulse width of 5 µs only, for readability. Secondly, the signals *V*_P_ for the microsecond pulses were compared, in the range 15 µs, with 1 µs steps. The results are reported in Figure 6a,b, respectively.

Figure 6a highlights that the shunt voltage has a short delay with respect to the trigger signal, and a relevant noise level is noted mainly in correspondence with the falling edge of the curve (close to 6 µs in the figure). The photodiode signal shows a larger delay with respect to the shunt signal, which is due to the transient response of the LED. The same delay can be found close to the falling edge. The photodiode signals were then compared in the case of pulses with increasing width. Figure 6b shows the results, normalized with respect to the maximum absolute value of the 5 µs pulse. As can be noted, all the curves share the same rising trend, reaching different maximum levels depending on the light power emitted by the LED.

Indirect camera measurements were then analyzed. In particular, the microsecond range was investigated by measuring pulses in the range of 1–5 µs, with 1 µs steps. Then the sub-microsecond operation was investigated in the range of 100–500 ns, with a 100 ns step, to assess the limitations of the proposed driver. It is worth noting that such short values are challenging in the field of mechanical vibration measurements, which are generally limited to 10 kHz, for which an exposure time of 2 µs was proven to be sufficient [25]. The experimental setup involved the LED, the digital camera, and the described photodiode sensor. The camera (Optomotive Smilodon, Ljubljana, Slovenia, 25 Mp, 99 fps at full resolution) was mounted to face the LED, and the lens (Computar VL6Z1626UC-MPYIR, 1.1” sensor, 16–96 mm varifocal) was set to the highest possible zoom. The pixels corresponding to the light emitting area were then selected in the image, cropping a portion of the image of approximately 203 × 259 pixels. The data from this emitting area were then measured over time and averaged to reduce the noise level. The experimental setup is shown in Figure 7a, while an example image and the LED emitting area are emphasized in Figure 7b.

A different lens aperture was adopted to measure the two families of pulses: fully open for the nanosecond pulses and fully closed for the microsecond pulses. This was needed to guarantee a proper acquisition of the short-width pulses and to avoid overexposure for the longer pulses. The camera frame rate was set at 50 Hz, while the LED pulses were set to have a frequency shift of 10^−5^ Hz, so that Δ*t* = 4·10^−9^ s was achieved (following Equation (4)). The results obtained through the indirect camera measurements are reported in Figure 8. Figure 8a refers to the microseconds range, while Figure 8b refers to the nanoseconds range.

All the pulses were normalized with respect to the absolute measured maximum, i.e., the response corresponding to a 5 µs pulse. Since the aperture was changed between the two pulse families, the pulse at 500 ns was measured with both apertures, and the results were used to adjust the scales to the same normalization. As can be noted from Figure 8, despite the pulse width, all the light pulses share the same rising curve. Nevertheless, only the curves at 4 and 5 µs reach the maximum illumination level, meaning that the transient response of the LED requires more than 3 µs to rise from 0 to 100% illumination, thus confirming the findings of Figure 6b. The plot highlights that both the LED and the control circuit can handle incredibly short pulses since the desired peak profile is obtained down to 200 ns. A response at 100 ns was also obtained, but the noise level was too high to retrieve precise information about the pulse shape since synchronization between LED and camera exposure requires too strict specification to retrieve such short light pulses.

Finally, a comparison between the direct and indirect pulse measurements was performed, i.e., photodiode and camera measurements, respectively. It is worth noting that a direct comparison between the indirect camera measurement and the photodiode signal is not trivial because the two measurements refer to different quantities: the former is the average of a large area of pixels over time, while the latter is a punctual measurement of a single sensor, so that the measured trends cannot be perfectly overlapped. Nevertheless, the signal measured by the photodiode (Figure 6b) was compared with the pulse reconstructed by the indirect camera approach (Figure 8). The sign of the photodiode signal was reversed to be comparable with the camera measurement, and each of the two signals was normalized with respect to its own maximum to have a fair comparison. The results are reported in Figure 9 for the 5 µs pulse (other pulses were omitted for readability).

The figure clearly shows a good agreement between the two independent measurements, both in terms of rise transient behavior and pulse duration. A relevant discrepancy can be noted in terms of fall transient behavior: even if both photodiode and camera signals appear to decrease after the expected timing (i.e., 5 µs), the former is considerably slower than the latter. This can be attributed to the low-cost hardware selected for the simple photodiode circuit, which could not guarantee the needed fall time for the sensor to properly follow the LED light pulse transient behavior. This discrepancy can be considered acceptable, considering the simple and cost-effective electronic configuration adopted for the photodiode setup, but it confirms that the camera-based approach can be more convenient since it does not require any additional hardware but is still more effective than the photodiode approach.

## 4. Conclusions

In this paper, a method to produce short-time high-power light pulses in the range of microseconds for Digital Image Correlation vibration measurements is introduced. The circuit schematic is discussed, providing large flexibility in light power control through an analog DC signal and in pulse timing through a TTL control signal. The possibility of retrieving feedback from the driving circuit is also discussed, both in terms of high-fidelity real-time LED shunt voltage measurement through an oscilloscope and of indirect estimation of the average LED current level through low sampling rate acquisition boards. The developed driver overcame the limitations of literature devices providing analogic control of the LED output power through a DC control signal and an analogic output that can be measured with a slow sampling rate (i.e., 100 Hz) to retrieve information about the actual current flowing through the LED without needing an oscilloscope with a high sampling frequency in the range of MHz. The experimental activity demonstrated that the developed driving board can produce light pulses in the range of microseconds and could even be pushed to a limit of a 100 ns light pulse.

Besides the LED driving circuit, two different low-cost methods are presented to assess the pulse width characteristic. Firstly, a photodiode sensor is designed based on off-the-shelf, low-cost components and is able to directly measure the high-speed light pulses emitted by the LED. Secondly, an innovative indirect measurement approach based on a low-speed digital camera is presented. By exploiting frequency modulation, it was possible to describe the LED transient response down to the nanosecond scale, even if the camera sampling rate was set to 50 Hz. In particular, this latter approach resulted in being more effective and less noisy than the photodiode approach, and did not require any additional hardware since at least one digital camera is always present in any DIC system. The developed method exploits a downsampling approach, so that the camera specification is not a limitation for the possibility of describing the short-time pulse shape. The developed measurement systems allowed for the characterization of the tested LED, which showed a rise and fall time in the range of a few microseconds. This means that it can be used to produce light pulses in the microsecond range, but it reaches its maximum light intensity only if the pulse width exceeds 3 µs. Additionally, the experiments confirmed that, if the LED is driven in such a low duty-cycle regime (i.e., 0.01% when 2 µs pulses are emitted at 50 Hz), the manufacturer specifications can be exceeded to increase the lighting power without damaging the LED. In particular, the maximum current flowing through the LED during the tests was increased up to 60 A (i.e., twice the rated value) without any damage to the LED. Further research could be focused on the characterization of Chip On Board (COB) LEDs, which can reach even higher power levels, but their performance in the time domain is generally not provided by the manufacturer. Additionally, further development of the driving circuit could allow for the driving of more than one LED in parallel or series to reach higher light intensities.

## Figures and Tables

**Figure 1 sensors-24-04103-f001:**
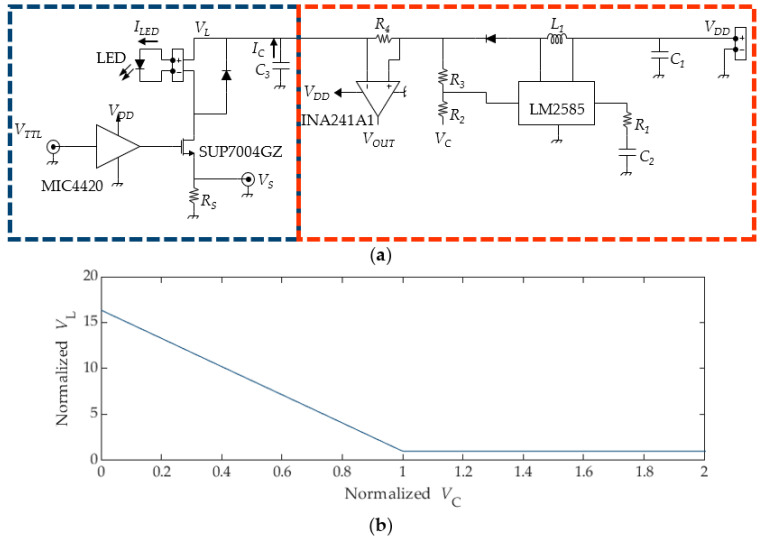
LED driving circuit: (**a**) direct LED driving circuit (blue box) and supply circuit, which controls the LED voltage and current (orange box), and (**b**) relation between *V*_C_ and *V*_L_.

**Figure 2 sensors-24-04103-f002:**
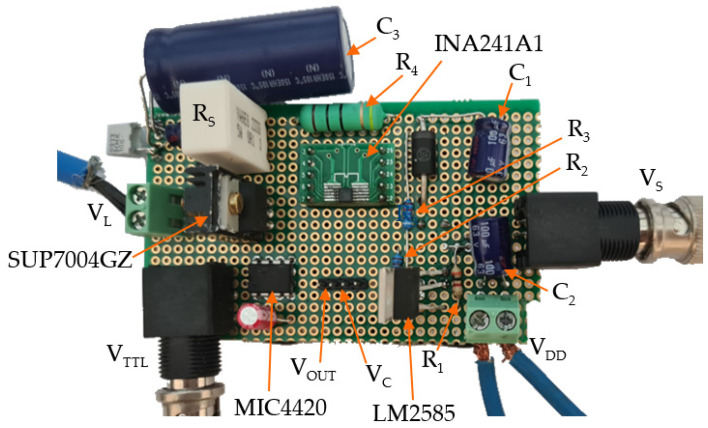
Assembled prototype of the described LED driver.

**Figure 3 sensors-24-04103-f003:**
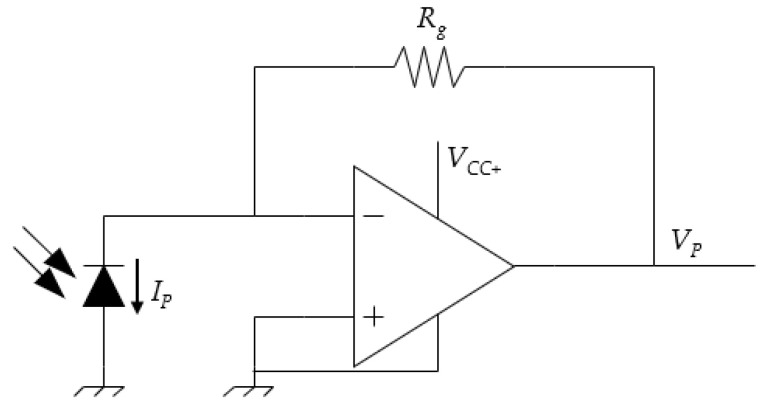
Circuit to read the high-speed photodiode signal.

**Figure 4 sensors-24-04103-f004:**
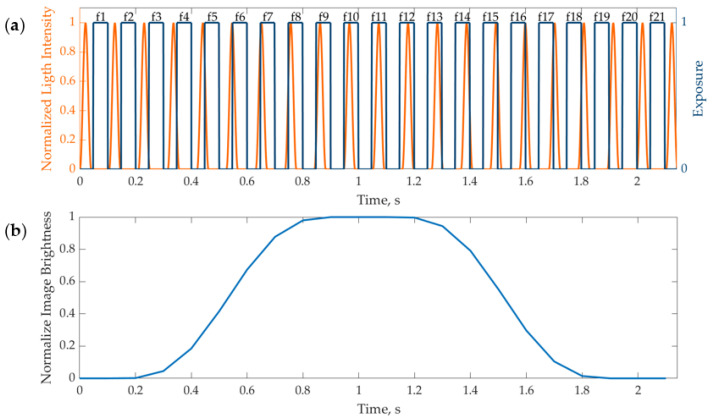
Numerical example of indirect camera acquisition of the light pulse: (**a**) time series of light pulses and camera exposure; and (**b**) average normalized image brightness over time.

**Figure 5 sensors-24-04103-f005:**
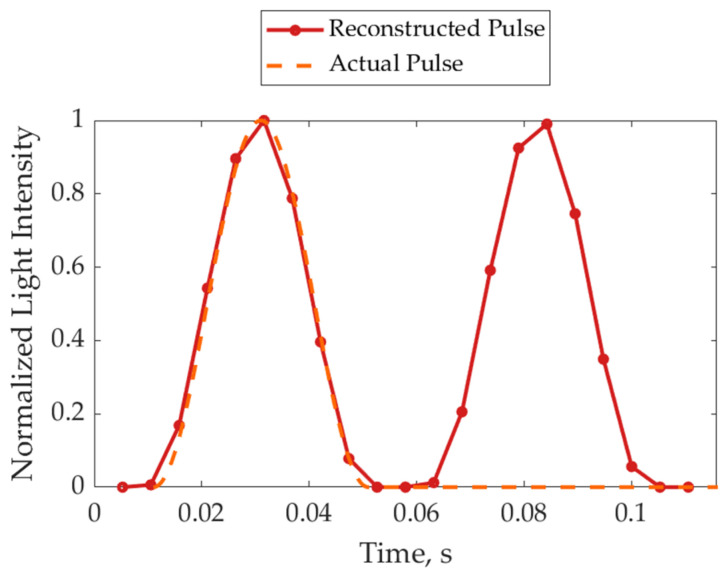
Numerical example of light pulse reconstruction through indirect camera measurements.

**Figure 6 sensors-24-04103-f006:**
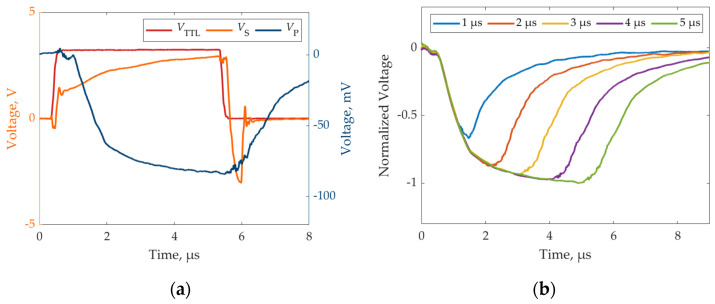
Experimental results of light pulses through direct photodiode measurements: (**a**) comparison of *V*_TTL_, *V*_S_, and *V*_P_, and (**b**) photodiode response for different pulse widths.

**Figure 7 sensors-24-04103-f007:**
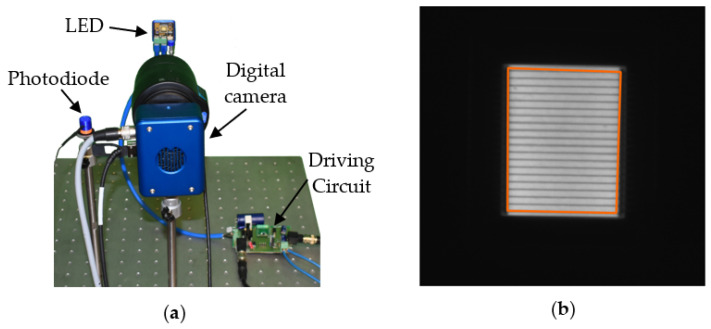
Experimental measurements: (**a**) setup and (**b**) light emitting area in camera images.

**Figure 8 sensors-24-04103-f008:**
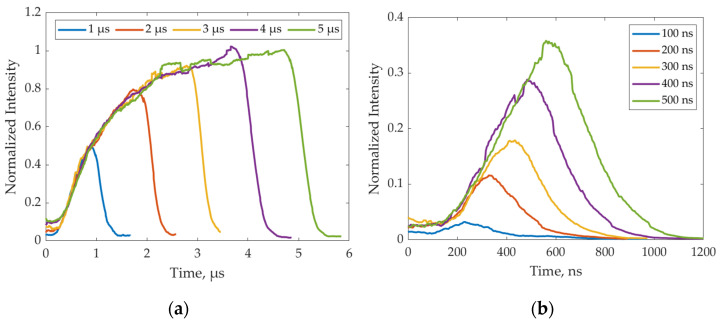
Experimental measurements of light pulses through the indirect camera method.

**Figure 9 sensors-24-04103-f009:**
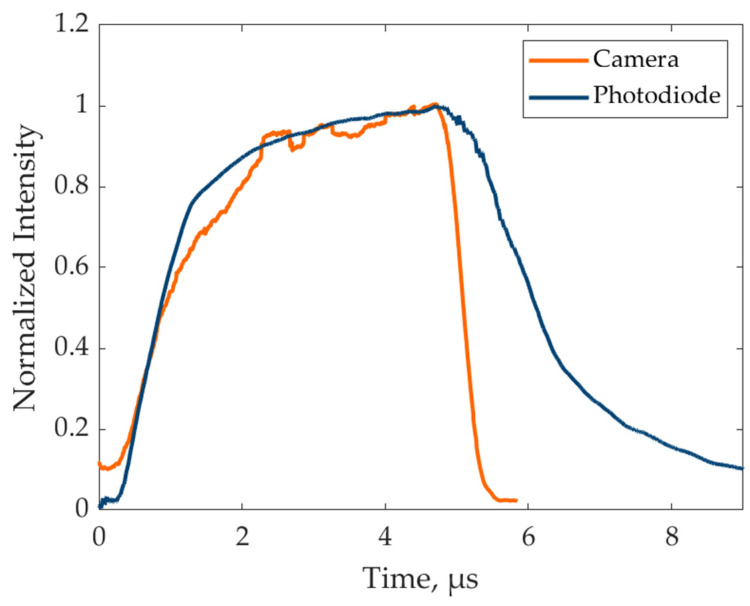
Comparison between direct and indirect measurement for the 5 µs light pulse.

## Data Availability

No new data were created.

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
