# Peer review of "High Power Pulsed LED Driver for Vibration Measurements"

_sensors, 2024, doi:10.3390/s24134103_

Round 1

Reviewer 1 Report

Comments and Suggestions for Authors

the high power pulsed LED driver is designed and the measurement can be clarified by the three different measurement ways. Especially the photodiode and digital camera are described well. In the introduction part, line 61 to 76, the reference number applied in the article directly is not so well understood by the readers. some more detailed description is suggested for this part.

Author Response

Reviewer #1

The high-power pulsed LED driver is designed and the measurement can be clarified by the three different measurement ways. Especially the photodiode and digital camera are described well.

The Author wishes to thank the Reviewer for the overall positive comments about the research, and for the rigorous and detailed revision which helped to improve the paper’s quality.

Comment 1:     In the introduction part, line 61 to 76, the reference number applied in the article directly is not so well understood by the readers. some more detailed description is suggested for this part.

Response:        The literature review was improved and the introduction was revised by adding more comments for each paper, as suggested.

Reviewer 2 Report

Comments and Suggestions for Authors

In this paper, a circuit is specifically designed to achieve high-intensity short-time light pulses, in the range of 1 µs. A prototype of the designed board was assembled and tested to check its capability to respect the specification. Three different measurement methods are proposed and validated to achieve short-time light pulse measurements: shunt voltage measurement, direct photodiode measurement with a low-cost sensor, and indirect pulse measurement through a low-frame-rate digital camera. This work will be of great interest to researchers in the field. I would like to give some comments and suggestions. The detailed comments are as follows:

1. In introduction, the authors write: “To this extent, the advancements in Light Emitting Diodes (LED) technology provided many accessible light sources that can be investigated [12]. The main drawback of such devices is that during continuous operation, the junction temperature can rise above admissible values, degrading the LED performance, shortening its life, or leading to abrupt failure [13,14]. To solve these issues, pulsed operation may be adopted, thus further pushing the maximum power of the LEDs.…” The general reference list in the introduction seems a bit thin, considering the evolution in the field within the recent years. To give the readers a much broader view, recent developments concerning on LEDs, such as Laser & Photonics Reviews 2024, 18, 2300464 (https://doi.org/10.1002/lpor.202300464); Optics Letters 47(5), 1291-1294 (2022) ; Nanoscale 14, 4887-4907 (2022), etc. should be added, so that the readers can be clear about the state-of-the-art of this topic.

2. Could the authors provide detailed information about the assembled prototype in figure 2?

3. In page 6, the authors indicate that camera delay was set to 48 ms, while camera and LED triggers were set to 10 Hz and 9.5 Hz, respectively. Why does the authors choose 0 Hz and 9.5 Hz? Is it possible to increase these values?

4. In figure 5, how did the author get the value of the actual light pulse?

5. The authors indicate that the LED investigated in this paper rates for 4.8 V and 30 A for continuous operation. Does it mean that the LED can work properly at 30 A current injection? As we can know, most LEDs cannot work in such a large current.

6. The Experimental setup in figure 7a is a bit puzzling. It is recommended to add arrows to illustrate each component.

Author Response

Reviewer #2

In this paper, a circuit is specifically designed to achieve high-intensity short-time light pulses, in the range of 1 µs. A prototype of the designed board was assembled and tested to check its capability to respect the specification. Three different measurement methods are proposed and validated to achieve short-time light pulse measurements: shunt voltage measurement, direct photodiode measurement with a low-cost sensor, and indirect pulse measurement through a low-frame-rate digital camera. This work will be of great interest to researchers in the field. I would like to give some comments and suggestions. The detailed comments are as follows:

The Author wishes to thank the Reviewer for the overall positive comments about the research, and for the rigorous revision which helped to improve the paper’s quality.

Comment 1:     In introduction, the authors write: “To this extent, the advancements in Light Emitting Diodes (LED) technology provided many accessible light sources that can be investigated [12]. The main drawback of such devices is that during continuous operation, the junction temperature can rise above admissible values, degrading the LED performance, shortening its life, or leading to abrupt failure [13,14]. To solve these issues, pulsed operation may be adopted, thus further pushing the maximum power of the LEDs.…” The general reference list in the introduction seems a bit thin, considering the evolution in the field within the recent years. To give the readers a much broader view, recent developments concerning on LEDs, such as Laser & Photonics Reviews 2024, 18, 2300464 (https://doi.org/10.1002/lpor.202300464); Optics Letters 47(5), 1291-1294 (2022) ; Nanoscale 14, 4887-4907 (2022), etc. should be added, so that the readers can be clear about the state-of-the-art of this topic.

Response:        The Authors thank the Reviewer for the suggestion, the literature review was expanded and commented as suggested.

Comment 2:     Could the authors provide detailed information about the assembled prototype in figure 2?

Response:        A deeper description of the prototype board was added to the paper, along with the indication of the main components with reference to the scheme of Figure 1(a).

Comment 3:     In page 6, the authors indicate that camera delay was set to 48 ms, while camera and LED triggers were set to 10 Hz and 9.5 Hz, respectively. Why does the authors choose 0 Hz and 9.5 Hz? Is it possible to increase these values?

Response:        Figure 4 was drawn based on a numerical example, to exemplify the adopted procedure. To this extent, the properties of the example (e.g. camera and LED frequencies, pulse duration, etc.) were choses so that the plot would be readable with a reasonable scale. These values can be arbitrarily increased (as demonstrated in the experimental activity), but using the same values as for the experiment would lead to a hardly readable plot, reducing the effectiveness of the figure. This was better explained in the text.

Comment 4:     In figure 5, how did the author get the value of the actual light pulse?

Response:        Figure 5 is referred to the numerical example, thus the simulated light pulse was known (i.e., the orange line in Figure 4(a)). This was clarified in the paper.

Comment 5:     The authors indicate that the LED investigated in this paper rates for 4.8 V and 30 A for continuous operation. Does it mean that the LED can work properly at 30 A current injection? As we can know, most LEDs cannot work in such a large current

Response:        This information was retrieved from the manufacturer’s specifications, which were added to the paper in the reference list. Obviously, such severe current conditions and continuous operation would require high-performance cooling of the LED.

Comment 6:     The Experimental setup in figure 7a is a bit puzzling. It is recommended to add arrows to illustrate each component.

Response:        Figure 7a was improved by adding arrows to highlight the different components.

Reviewer 3 Report

Comments and Suggestions for Authors

Neri et al. presented a cheap and effective method to fabricate and characterize an electrically-controlled high-intensity pulsed LED, which can be used for the imaging high-frequency oscillation. They designed the LED circuit, and experimentally fabricated it which they then characterized using three distinct methods. They confirmed the consistency across the various methods. This work can contribute to any application which requires cost-effective fabrication and testing of pulsed LEDs. Therefore, I recommend its publication. However, I have several questions.

1. In Figure 5, how exactly is the pulse reconstructed from taking the derivative of the curve in Fig. 4b? How can one reconcile the two very different time scales (i.e., seconds versus 0.1 s)?

2. Can the authors comment a little bit on the performance (or other specs) of their homebuilt photodiode, compared to the commercial Thorlabs PDA10A?

Author Response

Reviewer #3

Neri et al. presented a cheap and effective method to fabricate and characterize an electrically-controlled high-intensity pulsed LED, which can be used for the imaging high-frequency oscillation. They designed the LED circuit, and experimentally fabricated it which they then characterized using three distinct methods. They confirmed the consistency across the various methods. This work can contribute to any application which requires cost-effective fabrication and testing of pulsed LEDs. Therefore, I recommend its publication. However, I have several questions.

The Author wishes to thank the Reviewer for the overall positive comments about the research, and for the rigorous and detailed revision which helped to improve the paper’s quality.

Comment 1:     In Figure 5, how exactly is the pulse reconstructed from taking the derivative of the curve in Fig. 4b? How can one reconcile the two very different time scales (i.e., seconds versus 0.1 s)?

Response:        Each point in Figure 4(b) represents the intensity level of a frame acquired by the camera. This intensity is the integral of the light pulse, in the time window when the light pulse overlaps with the exposure of the camera (since exposure is a square wave in the range 0-1). Thus, it is possible to retrieve the light pulse shape from the data reported in Figure 4(b) by deriving the curve.

Concerning the time scales, the horizontal axis of Figure 4(b) reports the frames’ timestamps, which are slow. Nevertheless, in each frame a different time shift Δt between the light pulse and the exposure is introduced (equation 4). The value of Δt can be arbitrarily low, depending on the values of fc and fl. Thus, it is possible to assign to the i-th frame acquired by the camera the fictitious time instant i × Δt, so that when the derivative is computed, the fictitious time scale can be used to plot the data because the time shift of each frame is known.

Both these aspects were numerically and experimentally verified. These explanations were improved in the paper.

Comment 2:     Can the authors comment a little bit on the performance (or other specs) of their homebuilt photodiode, compared to the commercial Thorlabs PDA10A?

Response:        The Thorlabs PDA10A is rated up to 150 MHz, which corresponds to about 6.7 ns, thus the response time is in the same order of magnitude as the developed sensor. These comments were added to the paper.